# A Fuzzy-Based Co-Incentive Trust Evaluation Scheme for Edge Computing in CEEC Environment

**Geming Xia, Chaodong Yu \*** and **Jian Chen**

College of Computer Science and Technology, National University of Defense Technology, Changsha 410003, China
* Correspondence: chaodongyu16@nudt.edu.cn

**Abstract:** With the development of 5G and artificial intelligence, the security of Cloud-Edge-End Collaboration (CEEC) networks becomes an increasingly prominent issue due to the complexity of the environment, real-time variability and diversity of edge devices in CEEC networks. In this paper, we design a lightweight fuzzy collaborative trust evaluation model (LFCTEM) for edge devices, and calculate the trust values of edge devices by fuzzifying trust factors. To alleviate the selfish behavior of edge devices, this paper introduces an incentive mechanism in the trust evaluation model, and achieves a long-term incentive effect by designing an incentive negative decay mechanism, which enhances the initiative of collaboration and improves the interference resistance of CEEC networks. We verify the performance of LFCTEM through simulation experiments. Compared with other methods, our model enhances the detection rate of malicious edge devices by 19.11%, which improves the reliability of the CEEC trust environment. Meanwhile, our model reduces the error detection rate of edge devices by 16.20%, thus alleviating error reporting of the CEEC trust environment.

**Keywords:** edge computing; Cloud-Edge-End Collaboration; trust evaluation; fuzzy logic; incentive mechanism



## 1. Introduction

With the development of 5G and artificial intelligence, data generation is exploding, and the total amount of global data will grow to 175 ZB by 2025 [1]. More than 49% of these data will be stored in public cloud environments. These put forward extremely high requirements on network delay, data security and controllability [2]. Due to the huge amount of data contributed by massive IoT devices, traditional cloud processing produces high end-to-end delays and brings huge loads to the transmission communication network. In order to effectively meet the new requirements of low delay, privacy and energy saving, edge computing was born [3,4]. However, the limited computing power of edge end cannot fully meet the increasing demand of edge devices. Edge computing cannot replace cloud computing, but can act as a supplement to cloud computing together with edge devices, presenting an architecture form of Cloud-Edge-End Collaboration, as shown in Figure 1. However, the architecture of CEEC also has some security challenges. Firstly, because of the heterogeneity and complexity of CEEC networks, the traditional centralized security mechanism cannot be applied well. Secondly, due to the openness of CEEC networks and the high dynamic nature of edge devices, it is difficult to effectively identify malicious edge devices. However, edge devices have limited resources, which makes them vulnerable to attacks, and lack effective lightweight security mechanisms. Therefore, it is necessary to build a trusted environment and study the trust management of CEEC.

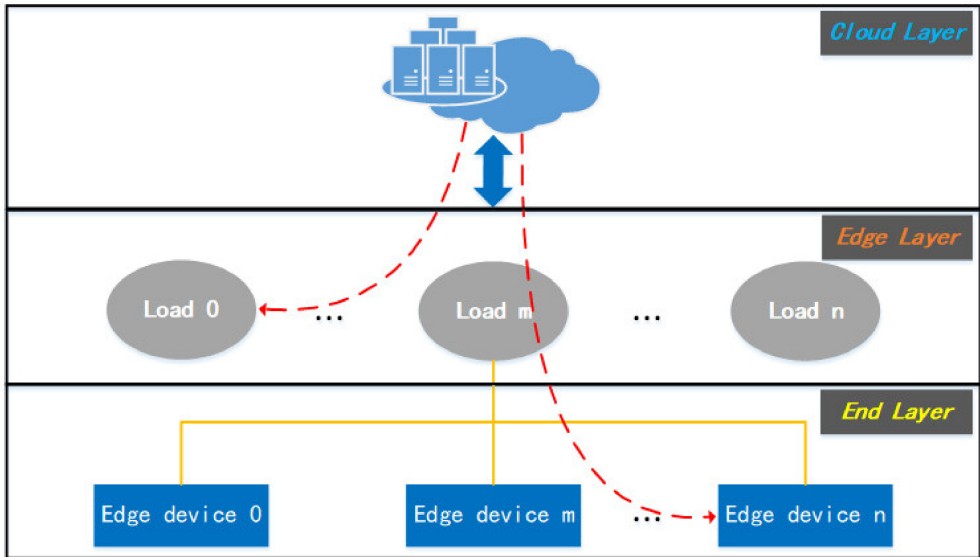

**Figure 1.** Cloud-Edge-End Collaboration architecture.

Some research has been conducted on trust evaluation in edge computing [5–7]. Yu et al. [8] took multi-dimensional trust data as the training set of a BP neural network to evaluate the trust degree of edge nodes in a computational power network, and adopted the improved particle swarm optimization algorithm to optimize the structure and weight of the neural network so as to effectively improve the detection rate of malicious nodes. This method adaptively adjusted the weight of trust aggregation. Particle swarm optimization was used as feedback to realize the incentive mechanism for edge nodes. However, it did not take into account privacy protection and computational costs. It was also not tested for anti-aggression. In the literature [9], the decision tree classification model method was used to construct trust rules, and the Euclidean distance concept was used to calculate the trust value in the calculation of recommendation trust. Furthermore, the artificial neural network function was used to self-train the vehicle nodes that did not meet the expected trust value to improve the vehicle trust degree. This work developed a manual training model as an incentive mechanism. However, it did not consider privacy protection and anti-aggression. It also did not take into account computational costs. In the literature [10], a capsule neural network was used to predict trust attributes based on historical data to obtain the trust value of edge devices. Only the shallow network structure and a small amount of training data were needed to realize the trust evaluation, which met the time requirements of frequent processing of multi-modal and small-sample data in edge scenarios. This study used a capsule neural network to aggregate trust attributes and overcome the subjectivity of trust weights. However, it did not consider privacy protection and incentives. It also did not analyze adaptability to attacks and did not consider the computational cost of the algorithm. Zhang et al. [11] proposed a credible edge platform by combining blockchain with edge computing. This realizes the lightweight design of the platform through the microservice architecture. It improves the portability of the platform by introducing the Edgex Foundry framework, and it also verifies the availability of the trusted edge platform by deploying it on multiple network nodes for simulation.

Huang et al. [12] used multi-weight subjective logic to realize the credit value calculation of the reputation system and construct the local and global reputation. They determined the weight of reputation according to the three aspects of familiarity, similarity and timeliness, and constantly updated the value of reputation. This method improved the efficiency of resource allocation. However, it did not take into account privacy protection and robustness against attacks. It also did not take into account inter-transmission incentives. At the same time, the establishment of weight in reputation calculation was subjective, which reduced the accuracy of trust values. Wang et al. [13] introduced trust

chains when evaluating trust of sensor nodes. The trust value of a single atomic trust chain combined interaction trust, energy trust and recommendation trust. The fine-grained trust values of sensor nodes were evaluated by serial and parallel trust transitions in the merged trust chain. This approach was robust against malicious attacks. However, it did not consider privacy protection and incentive of trust data. The computational cost of the algorithm has not been evaluated. It also did not discuss the weight of trust aggregation, which reduced the accuracy of trust values. In the literature [14], node trust was evaluated according to several parameters. Direct trust evaluation considered node communication success rate, proximity, packet loss rate and residual energy. In the evaluation of indirect trust, recommendation trust with different weights came from the surrounding nodes. Finally, the trust was aggregated to obtain a multi-dimensional comprehensive trust value. This method could resist malicious attacks and balanced the energy consumption of nodes. However, it did not consider privacy protection and incentives. In the calculation and aggregation of direct and indirect trust, it did not analyze the specific allocation of trust weight. Tian et al. [15] collected trust values from multi-source evaluation from the perception level, and assigned weights to different types of trust evaluation values according to service attribute sets, self-service evaluation factors, user quality evaluation factors and social sensor personality preferences so as to achieve multi-dimensional aggregation of trust values. This method considered the adaptability of trust weight and reduced the computational cost. However, it did not consider privacy protection and robustness against attacks. It also did not consider the incentive mechanism between sensor nodes.

Fuzzy logic constructs multiple fuzzy factors by modeling the uncertain factors, deducing the uncertainty relation of entity behavior through fuzzy operators, evaluating the subject using the fuzzy comprehensive evaluation algorithm and, finally, obtaining the result. Some examples from the literature [16–21] introduce fuzzy logic technology to implement trust management in edge computing. In the literature [16], prior knowledge in trust evaluation was provided by a fuzzy inference system, which combined expert knowledge of trust evaluation into a fuzzy rule base using fuzzy methods. It was also based on artificial neural networks to capture different patterns to make decisions about the trust status of IoT nodes. In the literature [17], the uncertainty in trust evaluation was divided into three fuzzy sets: packet loss factor, false packet injection factor and content change factor. The algorithm based on fuzzy logic was used to evaluate the trust degree of vehicles, and the network topology factor was introduced to combat data change attack. This method could resist packet loss attack, error injection attack and data change attack. It also solved the uncertainty of trust weight by aggregating trust factors through fuzzy logic. However, it did not consider privacy or incentives. It also did not estimate the computational cost. Serin et al. [18] reduced security threats by establishing a credible environment based on ambiguity in smart cities. The model could effectively avoid and isolate malicious nodes in IoT, and reduced the impact of collusion attacks. However, the above work is one-sided in the extraction of trust data when constructing fuzzy factor subsets, which will affect the accuracy of trust evaluation values. In order to improve service quality, Hossain et al. [19] designed the collaborative task offloading scheme FCTO by taking the delay sensitivity of QOS as the fuzzy input parameter. This scheme optimizes the task completion time and server utilization by introducing fuzzy logic in task offloading.

Due to the selfish behavior of edge nodes, the quality and reliability of their trust evaluation in complex and changeable network environments are affected. An incentive mechanism can be introduced to reward the quality of trust evaluation results so as to improve the initiative of edge nodes in trust evaluation with limited resources. The incentive mechanism in trust evaluation is the direction of future research. In recent years, some research has taken incentive into account. In the literature [22], smart contracts were used to automatically punish the misconduct of stakeholders. Additionally, decentralized accountability mechanisms and automatic rewards were developed to encourage nodes to verify collaboration services and provide verification rewards. This regulated the behavior of participants and built a reliable edge computing network. In the literature [23], the cloud

server set rewards for data collectors who actively collected sparse and high-quality data in remote areas so as to increase the collection contribution and, thus, improve the overall quality of data collection. In the literature [24], an incentive mechanism was designed: the winner of the competition was given the basic reward, and those who finished early were given extra rewards. These rewards would entice edge servers to complete computational tasks initiated by their peers and encouraged them to complete them quickly. In the literature [25], vehicles were encouraged to intentionally maintain trusted data sharing through the self-executing nature of smart contracts in a blockchain. It used the contribution of the vehicle as an incentive and adjusted the proportion of the reward according to how trustworthy the data were. In the literature [26], a penalty factor was introduced into the trust value of the vehicle, so that the vehicle refusing to provide service can obtain a lower trust value, and if the vehicle provided a false location, its trust value would be greatly reduced. This discouraged dishonest vehicles and incentivized selfish vehicles to actively offer their services. In the literature [27], users were motivated by the development of an effective voting mechanism with built-in consensus and punishment functions in the blockchain. The nodes involved in the verification would be rewarded with part of the deposit, while the abusive users would be punished and blacklisted. In the literature [28], UAVs were used to collect partially verifiable data, and random sampling by trusted third parties was used for joint trust evaluation. Therefore, the uncertainty of verifiable collected data increased the likelihood of detecting fake data uploads by mobile edge users, thus achieving the purpose of motivating users to upload trusted data honestly. In the literature [29], when evaluating trust in data in federated learning, a trust reward and punishment method was proposed in order to achieve the incentive of consensus of trust. When a task was completed, a certain amount of credit would be added. However, when the trust value was large, the proportion of increased trust value would be reduced so that the trust value of nodes added later could quickly catch up. In order to realize the fairness among all nodes, the punishment degree of each node was different. The higher the trust value, the harsher the punishment was to ensure the fairness of rewards and punishments. However, all the above research works were short-term incentives for the edge devices, and the effect of incentives was limited, which may cause the intermittent selfish behavior of the edge devices.

Overall, our study investigates the gaps in the above studies and makes the following contributions:

- Firstly, we propose a Cloud-Edge-End Collaboration (CEEC) computing power architecture and design a lightweight fuzzy cooperative trust evaluation model (LFCTEM) for edge devices, which fuzzies the uncertainty factors in the CEEC network. By constructing four fuzzy trust factors, we evaluate the trust value of edge devices according to the success rate of direct interaction, adjacent distance, public interaction success rate and cooperative positivity of the edge device group.
- Secondly, we introduce the incentive mechanism in the trust evaluation algorithm to encourage the edge devices through the cooperation reward scores, quality reward scores and efficiency reward scores. We also adopt the incentive negative decay mechanism to improve the continuity of active cooperation of edge devices so as to avoid their intermittent selfish behavior.
- Finally, we present experiments carried out on the OMNET++5.6.2 simulation platform, and further comparison of our algorithm with DTEM and FLTEEV algorithms, thus proving the effectiveness and advantages of our model.

## 2. Materials and Methods

### 2.1. Cloud-Edge-End Collaboration (CEEC) Computing Power Architecture

The architecture of each cooperative computing power in the CEEC environment is shown in Figure 2. The cooperative computing system in the CEEC architecture includes the cloud layer, edge layer and end layer.

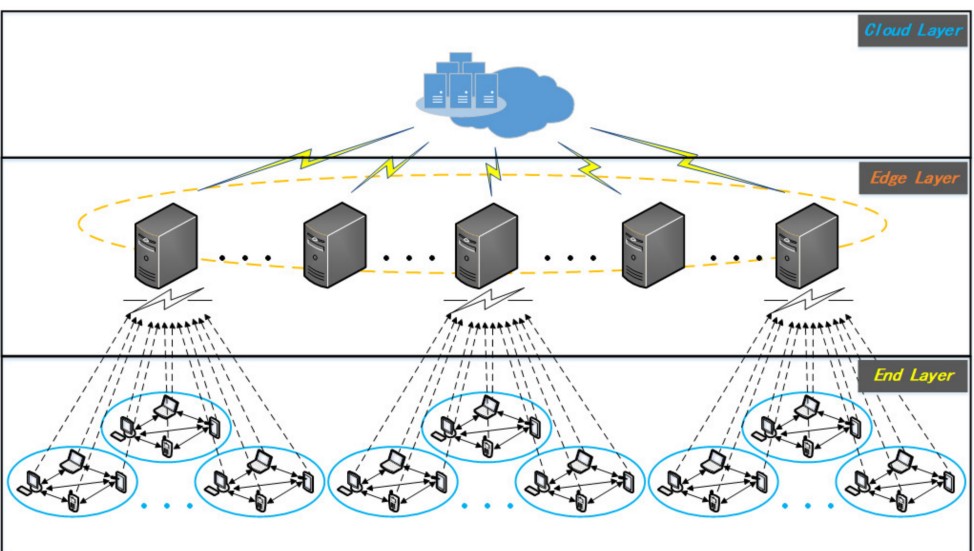

**Figure 2.** Cloud-Edge-End Collaboration (CEEC) computing power architecture.

The cloud layer has high computing power, and the powerful computing resources of the cloud can be used to achieve the incentive reward for the edge devices. When the edge devices have intermittent selfish behavior, its incentive integral is negatively attenuated.

The edge layer includes edge servers. Each edge server stores the adjacent history interaction data and incentive scores of dynamic heterogeneous computing powers in the region. Edge servers cooperate to evaluate the trust value between edges and manage the global trust database so as to ensure the efficient operation of CEEC networks.

The end layer is mainly composed of a variety of heterogeneous computing devices at the edge, including intelligent sensors, desktops, intelligent vehicles, intelligent gateways and so on. Each dynamic heterogeneous computing device is abstracted as a dynamic heterogeneous edge device. Because these edge devices are dynamically heterogeneous, with a change in their position they will enter the area managed by different edge servers and the dynamic heterogeneous computing devices in this area will perform collaborative computing within the edge.

### 2.2. Associated Trust and Security Issues

There are several trust and security issues in CEEC networks as follows:

Privacy protection issues: When maintaining trust in edge computing, users need to exchange trust data frequently, which may contain their private information. Disclosure of private data will create additional security concerns and most research does not take privacy into account. With the continuous promotion of edge computing and the rapid growth of IoT devices, the problem of data privacy warrants further study. Privacy protection models also need to adapt to the needs of various complex edge computing scenarios.

Trust weight allocation issues: In the trust evaluation methods based on traditional mathematical statistics, the trust value is usually determined by aggregating trust factors through weighting and other related calculations. However, there is no basis to determine the trust weight. Some studies determine the trust weight based on subjective judgment, while others determine the trust weight based on experience. All of these will lead to inaccurate trust weight allocation, thus affecting the results of trust evaluation. Therefore, it is necessary to further study the allocation of trust weight and develop a new trust aggregation method.

Edge intelligence trust management problem: In the edge intelligence scenario, there is no trust management mechanism for heterogeneous data owners. In edge intelligence networks, a single data owner may hold multiple training data sets that are heterogeneous in quality and type. The existing trust model is mainly oriented to a single application scenario and cannot realize the compatible trust management of multiple heterogeneous training

data sets held by a single data owner. There is a lack of unified management mechanism to fully and flexibly manage the trust of the data owners of the edge intelligence networks.

*2.3. Lightweight Fuzzy Collaborative Trust Evaluation Model*

In order to effectively evaluate the trust value of edge devices in the CEEC environment, we design a lightweight fuzzy cooperative trust evaluation model (LFCTEM). In this paper, four trust factors generated by fuzzy sets are used to fully model the neighboring cooperation attributes of edge devices and the trust value of edge devices is evaluated based on the fuzzy logic algorithm. The fuzzy logic system is used to overcome the indirect uncertainty of the trust mechanism.

In the CEEC environment, a group of neighboring edge devices is used to evaluate the trust value of one of the edge devices so as to further improve the reliability of the trust evaluation value. However, due to the complexity of the CEEC environment, the unreliability of wireless communication transmission and the unpredictability of edge device behavior, uncertainty will be introduced in the collaboration trust evaluation. Uncertainty is also introduced by the fact that the edge devices often generate inaccurate and incomplete information. In the CEEC environment, any slight change may lead to mismatch between the calculation result of the trust value and the real-time state of the edge device, thus affecting the accuracy of the trust value. Therefore, in the continuous cycle, we use fuzzy logic to alleviate this mismatch. In this paper, the input–output relation of the fuzzy algorithm based on the IF–THEN rule is constructed.

The fuzzy logic algorithm designed in this paper includes four trust factors generated by fuzzy sets as input, which are direct interaction success rate factor (DISRF), adjacent distance factor (ADF), public interaction success rate factor (PISRF) and cooperative incentive mechanism factor (CMF). Their membership functions are shown in Figure 3.

(1)   Direct interaction success rate factor (DISRF) represents the interaction success rate between an edge device and its neighboring edge devices. It is an entity-centric trust factor. Due to the high dynamic of edge devices and the unreliability of wireless communication transmission, the number of neighboring edge devices is constantly changing, which leads to the uncertainty of their direct trust relationship. With the decrease in the success rate of direct interaction, the uncertainty of the direct trust relationship with neighboring edge devices will increase. The calculation of the interaction success rate between an edge device and its neighboring edge devices is shown in Equation (1). In this paper, DISRF is set to include two options: malicious and normal, ranging from 0 to 100%. The membership function of DISRF is shown in Figure 3a.

(2)   Adjacent distance factor (ADF) represents the interaction distance between the host and guest of an edge device group. It is an entity-centric trust factor. Because the density and environment of the edge device group will change over time, the recommendation weight of the edge device group is uncertain when evaluating the collaborative recommendation trust. At the same time, recommended trust is mainly obtained based on the indirect trust relationship between the public edge devices of the host and guest edge devices. As the adjacent distance between the host and guest edge devices increases, the uncertainty of its indirect trust relationship with the public edge devices will also increase. The calculation of the interaction distance between the host and guest of edge device group is shown in Equation (2). In this paper, ADF is set to include three options: short, medium and long, ranging from 0 to 300 m. The membership function of ADF is shown in Figure 3b.

(3)   Public interaction success rate factor (PISRF) represents the interaction success rate between the host and guest of an edge device group and their public edge devices. It is an entity-centric trust factor. Due to the high dynamic of edge devices and the unreliability of wireless communication transmission, the number of public edge devices between the host and the guest edge devices is constantly changing, which leads to the uncertainty of their indirect trust relationship. With the decline in the

success rate of public interaction, the uncertainty of the indirect trust relationship with the public edge devices will also increase. The calculation of the interaction success rate between the host and guest of the edge device group and their public edge device is shown in Equation (3). In this paper, PISRF is set to include three options: low, medium and high, ranging from 0 to 100%. The membership function of PISRF is shown in Figure 3c.

(4)　Collaboration incentive mechanism factor (CMF) represents the positive inclination of neighboring edge device groups to participate in collaboration. It is a data-centric trust factor. Due to the selfishness of edge devices, they may intentionally limit the consumption of computing resources in order to protect their own interests during the collaborative tasks, which leads to the uncertainty of the collaborative behavior of edge devices. As the collaboration positivity of the edge devices decreases, the uncertainty of the recommendation trust value generated by the adjacent edge devices will also increase. The incentive score for the group of adjacent edge devices is described in detail in the next section. In this paper, CMF is set to include three options: negative, normal and active, ranging from 0 to 300. The membership function of CMF is shown in Figure 3d.

$$\varrho_{ij} = \frac{\sum_{\{Device_n\}} R_r}{\sum_{\{Device_n\}} R_r + R_e} \tag{1}$$

where $\varrho_{ij}$ is the interaction success rate between an edge device and its neighboring edge devices, $\{Device_n\}$ is the neighboring edge devices of the edge device, $R_r$ is the number of successful interactions and $R_e$ is the number of failed interactions.

$$d_{ij} = \sqrt{\left(d_i^x - d_j^x\right)^2 + \left(d_i^y - d_j^y\right)^2} \tag{2}$$

where $d_{ij}$ is the interaction distance between the host and guest of the edge device group, is the location of the host edge device, $d_i^x$ and $d_i^y$ is the location of the guest edge device.

$$\varrho_{mn} = \frac{\sum_{\{Device_k\}} R_r}{\sum_{\{Device_k\}} R_r + R_e} \tag{3}$$

where $\varrho_{mn}$ is the interaction success rate between the host and guest of the edge device group and their public edge devices, $\{Device_k\}$ is the public edge devices for the host and guest of the edge device group, $R_r$ is the number of successful interactions and $R_e$ is the number of failed interactions.

The output of the fuzzy logic algorithm designed in this paper is the trust evaluation value (TEV), which contains four options: Untrust, Trust-3, Trust-2 and Trust-1, ranging from 0 to 1. The membership function of TEV is shown in Figure 4. The fuzzy logic algorithm consists of 54 fuzzy logic rules, as shown in Table 1.

When the output membership value of the fuzzy logic algorithm is obtained, it will be defuzzed to obtain the trust evaluation value. In this paper, the fuzzy logic library of MATLAB is used to construct the fuzzy logic trust evaluation algorithm and deblur the output membership value.

### 2.4. Incentive Score Mechanism

Since all the participants in the trust evaluation are dynamically distributed and organized together, and their management is relatively loose, the incentive mechanism can effectively ensure the quality of the collaborative evaluation. For the edge devices, the benefits of participating in the trust evaluation should be proportional to the computing resources invested so as to keep their positivity towards continuing to participate in the collaborative evaluation. For the edge side, the quality of the evaluation should be improved by recruiting high-quality edge devices for collaboration through incentive mechanisms.

Therefore, we need to find a balance between the consumption of edge computing resources and the quality of collaborative trust evaluation. We designed an incentive mechanism and introduced incentive scores. The incentive score can be used by edge devices to obtain resources and collaboration assistance, thus ensuring the integral value of the incentive score in an edge collaboration environment. The key idea is to stimulate the positivity of the edge devices towards participating in the cooperative trust evaluation through incentive scores, and to use negative attenuation to improve the persistence of the positive cooperation of the edge devices.

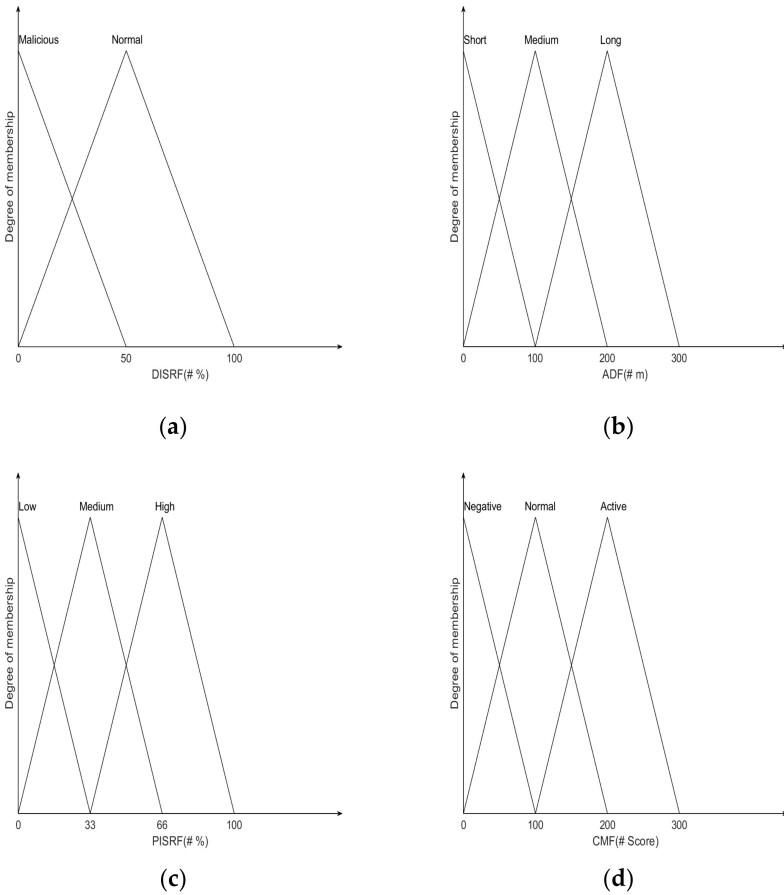

**Figure 3.** The input membership function of LFCTEM: (**a**) direct interaction success rate factor; (**b**) adjacent distance factor; (**c**) public interaction success rate factor; (**d**) cooperative incentive mechanism factor.

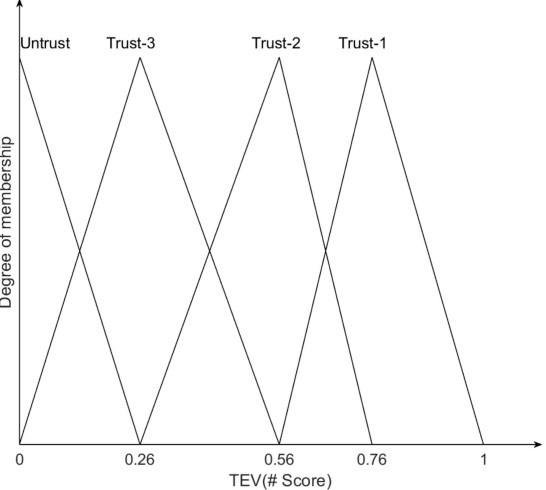

**Figure 4.** The output membership function of LFCTEM.

**Table 1.** Fuzzy logic rule.

| No. | DISRF | ADF | PISRF | CMF | TEV |
|-----|-------|-----|-------|-----|-----|
| 1 | Malicious | Short | Low | Negative | Untrust |
| 2 | Malicious | Short | Medium | Negative | Untrust |
| 3 | Malicious | Short | High | Negative | Untrust |
| 4 | Malicious | Short | Low | Normal | Untrust |
| 5 | Malicious | Short | Medium | Normal | Untrust |
| 6 | Malicious | Short | High | Normal | Trust-3 |
| 7 | Malicious | Short | Low | Active | Untrust |
| 8 | Malicious | Short | Medium | Active | Trust-3 |
| 9 | Malicious | Short | High | Active | Trust-3 |
| 10 | Malicious | Medium | Low | Negative | Untrust |
| 11 | Malicious | Medium | Medium | Negative | Untrust |
| 12 | Malicious | Medium | High | Negative | Trust-3 |
| 13 | Malicious | Medium | Low | Normal | Untrust |
| 14 | Malicious | Medium | Medium | Normal | Untrust |
| 15 | Malicious | Medium | High | Normal | Trust-2 |
| 16 | Malicious | Medium | Low | Active | Untrust |
| 17 | Malicious | Medium | Medium | Active | Trust-3 |
| 18 | Malicious | Medium | High | Active | Trust-2 |
| 19 | Malicious | Long | Low | Negative | Untrust |
| 20 | Malicious | Long | Medium | Negative | Trust-3 |
| 21 | Malicious | Long | High | Negative | Trust-3 |
| 22 | Malicious | Long | Low | Normal | Untrust |
| 23 | Malicious | Long | Medium | Normal | Trust-3 |
| 24 | Malicious | Long | High | Normal | Trust-2 |
| 25 | Malicious | Long | Low | Active | Untrust |
| 26 | Malicious | Long | Medium | Active | Trust-2 |
| 27 | Malicious | Long | High | Active | Trust-2 |
| 28 | Normal | Short | Low | Negative | Trust-3 |
| 29 | Normal | Short | Medium | Negative | Trust-3 |
| 30 | Normal | Short | High | Negative | Trust-3 |
| 31 | Normal | Short | Low | Normal | Trust-3 |
| 32 | Normal | Short | Medium | Normal | Trust-2 |
| 33 | Normal | Short | High | Normal | Trust-1 |
| 34 | Normal | Short | Low | Active | Trust-3 |
| 35 | Normal | Short | Medium | Active | Trust-1 |
| 36 | Normal | Short | High | Active | Trust-1 |
| 37 | Normal | Medium | Low | Negative | Trust-3 |
| 38 | Normal | Medium | Medium | Negative | Trust-2 |
| 39 | Normal | Medium | High | Negative | Trust-1 |
| 40 | Normal | Medium | Low | Normal | Trust-3 |
| 41 | Normal | Medium | Medium | Normal | Trust-2 |
| 42 | Normal | Medium | High | Normal | Trust-1 |
| 43 | Normal | Medium | Low | Active | Trust-2 |
| 44 | Normal | Medium | Medium | Active | Trust-1 |
| 45 | Normal | Medium | High | Active | Trust-1 |
| 46 | Normal | Long | Low | Negative | Trust-3 |
| 47 | Normal | Long | Medium | Negative | Trust-1 |
| 48 | Normal | Long | High | Negative | Trust-1 |
| 49 | Normal | Long | Low | Normal | Trust-2 |
| 50 | Normal | Long | Medium | Normal | Trust-1 |
| 51 | Normal | Long | High | Normal | Trust-1 |
| 52 | Normal | Long | Low | Active | Trust-2 |
| 53 | Normal | Long | Medium | Active | Trust-1 |
| 54 | Normal | Long | High | Active | Trust-1 |

In the CEEC environment, the incentive score mechanism is used to achieve the balance between resource regulation and collaborative evaluation. As shown in Figure 5, the edge devices can obtain the incentive score in the following three ways:

(1)  When the edge device is assigned a collaborative trust evaluation task and completes the evaluation task it is responsible for on time, it will receive the basic cooperation reward score.

(2)  According to the quality of the evaluation task assigned to the edge device, it will receive a quality reward score. In this paper, the completion quality of the trust evaluation task is evaluated mainly by the accuracy of collaborative evaluation. By setting the threshold, the completion quality is divided into three levels (excellent, medium and low), and the corresponding quality reward scores are matched according to the completion quality of different levels.

(3)  Based on the time it takes to complete the evaluation task assigned to the edge device, it receives an efficiency reward score. When the edge device completes the task assigned to it in advance, the corresponding efficiency reward scores are matched according to the time to complete the task ahead of schedule.

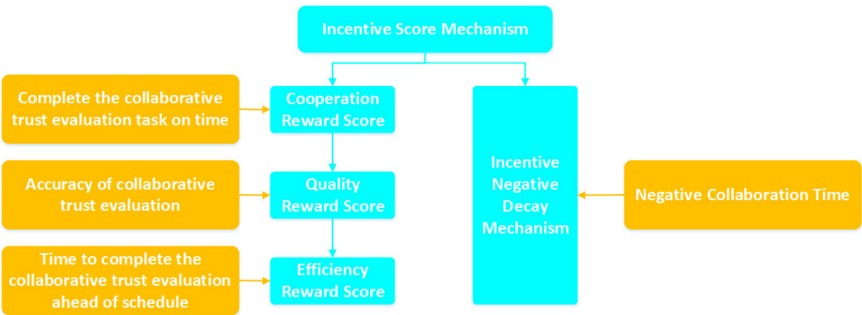

**Figure 5.** Edge cooperative incentive score mechanism.

When the edge initiates a cooperative trust evaluation task $T_{\{user\}}$ for the edge device group, the edge first broadcasts the information about the cooperative trust evaluation task $T_{\{user\}}$ on the CEEC environment, including the maximum completion time $\tau_{max}$ and the basic cooperative reward score $R_b$ of $T_{\{user\}}$. The value of $R_b$ is mainly determined by the amount of task computation and the priority of $T_{\{user\}}$ at the edge. For collaborative trust evaluation tasks with heavy computation or high priority, the basic cooperation reward scores are also higher, which improves the attraction of collaborative edge devices and facilitates the completion of collaborative trust evaluation tasks. When the collaborative trust evaluation task $T_{\{user\}}$ is completed, the total incentive score awarded to the collaborative edge devices are calculated as follows:

$$R = R_b + R_q + R_e \tag{4}$$

where $R_b$ is basic cooperation reward score, $R_q$ is quality reward score and $R_e$ is efficiency reward score.

$R_q$ is calculated from the quality of completion of the trust evaluation tasks assigned to the edge devices. We set the quality threshold $\varphi_1$ and $\varphi_2$ ($0 \leq \varphi_1 < \varphi_2 < 1$), and the completion quality is divided into excellent (collaborative evaluation quality is $\varphi_2$ or above), medium (collaborative evaluation quality is between $\varphi_1$ and $\varphi_2$) and low (collaborative evaluation quality is below $\varphi_1$). In order to motivate the edge devices to complete the cooperative trust evaluation task with high quality, we give the edge devices with excellent completion quality a higher increase in quality reward scores. For collaborative edge devices with moderate completion quality, we give moderate quality reward scores to maintain their base motivation. For collaborative edge devices with low completion quality, quality reward scores will not be given in this paper. The calculation of $R_q$ is as follows:

$$R_q = \begin{cases} \vartheta_q \cdot e^{(\varphi_q - \varphi_2)}, & \varphi_2 \leq \varphi_q \leq 1 \\ \frac{\vartheta_q \cdot (\varphi_q - \varphi_1)}{\varphi_q}, & \varphi_1 \leq \varphi_q < \varphi_2 \\ 0, & 0 \leq \varphi_q < \varphi_1 \end{cases} \tag{5}$$

where $\vartheta_q$ is the baseline quality reward score, $\varphi_1$ and $\varphi_2$ are the quality threshold of the collaborative trust evaluation task and $\varphi_q$ is the completion quality of the collaborative trust evaluation task.

$\varphi_q$ is calculated from the accuracy of collaborative evaluation:

$$\varphi_q = \frac{\delta_r}{\delta_r + \delta_e} \tag{6}$$

where $\delta_r$ is the number of times that the collaboration evaluation of the edge device group is correct and $\delta_e$ is the number of times that the collaboration evaluation of the edge device group is wrong.

$R_e$ is calculated from the completion time of the evaluation task assigned to the edge device:

$$R_e = \begin{cases} \frac{\vartheta_e \cdot (\tau_{max} - \tau_e)}{\tau_e} & , \quad \tau_e < \tau_{max} \\ 0 & , \quad \tau_e \geq \tau_{max} \end{cases} \tag{7}$$

where $\vartheta_e$ is the baseline efficiency reward score and $\tau_e$ is the actual time for the cooperative edge device to complete the collaboration trust evaluation task.

When the actual completion time of the collaborative edge device is shorter, the efficiency reward score will be higher. Conversely, when the actual completion time of the cooperative edge device exceeds $\tau_{max}$, it will not receive the efficiency reward score.

Through this incentive reward mechanism, CEEC will reward collaborative edge devices that can accomplish the task of collaborative trust evaluation quickly and with high quality. The reward score is mainly used as a short-term incentive to quickly motivate cooperating edge devices.

### 2.5. Incentive Negative Decay Mechanism

When the cooperative edge devices have a high incentive scores reserve, there is no urgent need for incentive scores in the short term. In this case, short-term collaboration negativity may occur on such collaborative edge devices, which will affect the efficient collaboration of the CEEC environment. Therefore, in order to achieve long-term incentive and maintain the continuity of the active cooperation of the edge devices, we designed the incentive negative decay mechanism.

The main idea of the incentive negative decay mechanism is as follows. In this paper, the decay time interval $\tau_d$ is set as the negative decay threshold. When the collaborative edge device is in a negative state for a long time (that is, the collaborative edge device is not participating in the collaboration evaluation for longer than $\tau_d$), its incentive score will decay significantly. In order to avoid the significant decay of incentive score, the cooperative edge device needs to continuously participate in the cooperative evaluation task or intermittently participate in the cooperative evaluation task in a short time interval. This will maintain the continuity of the active cooperation of the edge device and achieve the effect of long-term incentive.

In the incentive negative decay mechanism, in order to give larger incentives to the cooperative edge device, the incentive score will decay rapidly with the increase in time in the initial decay stage. This is to urge the edge device to participate in the cooperative trust evaluation task as soon as possible. In the mid-decay stage, the decay amplitude of the incentive score will be properly slowed down so as to give the cooperative edge device reaction time to obtain the incentive score again. In the late decay stage, if the cooperative edge device does not participate in the cooperative trust evaluation task when it reaches the maximum decay time $\tau_{dmax}$, its incentive score will decay to approach 0. Then the cooperative edge device will become a passive edge device and will not be able to access the computing resources and collaboration assistance provided by the CEEC environment.

Five combinations of undetermined formulas are considered in the design of the negative decay formula of incentive score in the incentive negative decay mechanism and we compare their decay rates comprehensively. The combinations of five undetermined

formulas are shown in Table 2, and their decay rates are shown in Figure 6. In this paper, the initial decay stage is set between 0 and 6 h, the mid-decay stage is set between 6 and 16 h and the late decay stage is set between 16 and 48 h. We set the maximum decay time $\tau_{dmax}$ to 48 h (two days). When the collaborative edge device does not participate in the collaborative trust evaluation task for two days, it is considered as a negative edge device and its incentive score decays to 0. According to Figure 6, it can be found that the attenuation ranges of $R_3$ and $R_5$ are too small, and they cannot approach 0 in the late decay stage. $R_4$ can approach 0 in the late decay stage, but its attenuation amplitude in the initial decay stage is not enough to give the cooperative edge device a large incentive. If the attenuation amplitude is too large in the initial decay stage, the incentive score will approach 0 in the mid-decay stage, and the mid-decay stage is not reserved for sufficient reaction time of the cooperative edge device.

**Table 2.** Incentive score negative attenuation formula.

| Decay Rate of Reward Scores (%) | Time (Hour) |
|---|---|
| $R_1$ | $\dfrac{R}{\left[\frac{5\cdot(\tau_n-\tau_d)}{72}+1\right]^3}$ |
| $R_2$ | $\dfrac{R}{\left[\frac{5\cdot(\tau_n-\tau_d)}{72}+1\right]^2}$ |
| $R_3$ | $\dfrac{R}{\log\left[\frac{5\cdot(\tau_n-\tau_d)}{72}+3\right]}$ |
| $R_4$ | $\dfrac{R}{e^{\frac{5\cdot(\tau_n-\tau_d)}{72}}}$ |
| $R_5$ | $\dfrac{R}{\frac{5\cdot(\tau_n-\tau_d)}{72}+1}$ |

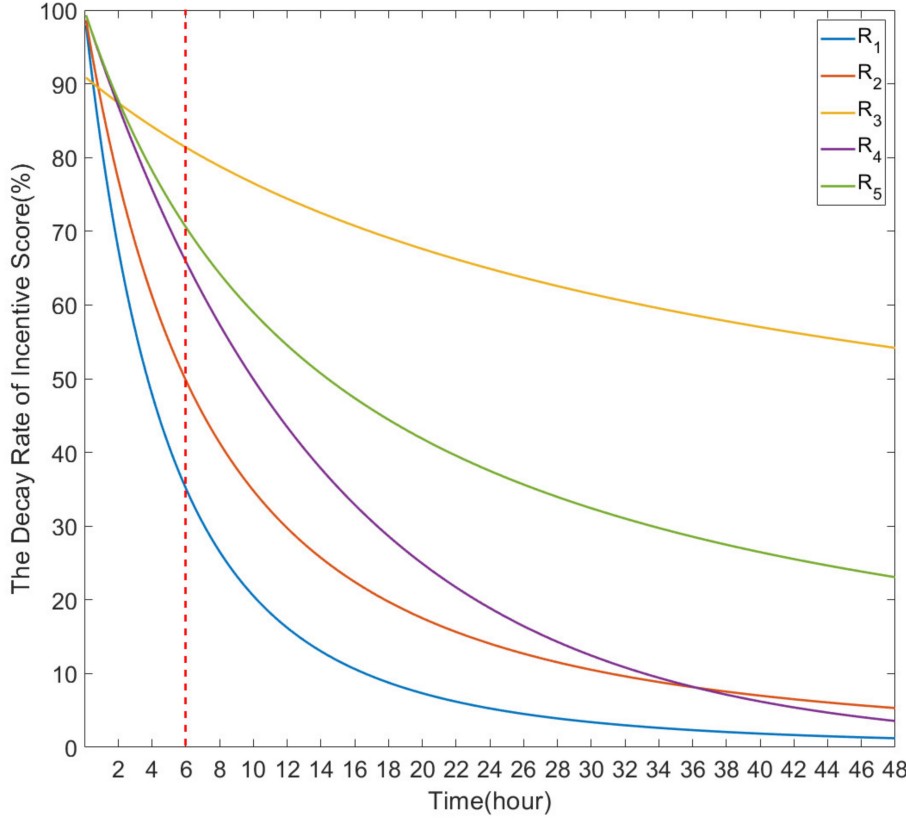

**Figure 6.** Comparison of negative decay of incentive score.

After comprehensive analysis, we found that the decay rate of incentive integral of $R_2$ reaches 50% at the initial decay stage, which gives the cooperative edge device a large incentive. In the mid-decay stage, the decay rate of the incentive score gradually decays

from 50% to 15%, giving the cooperative edge device sufficient reaction time to reacquire the incentive score. It can also approach 0 in the late decay stage, which meets the demand of the negative decay mechanism of incentive negative decay mechanism. Finally, we determined that $R_2$ is the negative decay formula of the incentive score.

Therefore, the calculation formula of the incentive negative decay mechanism is as follows:

$$R_d = \frac{R}{\left[\frac{5 \cdot (\tau_n - \tau_d)}{72} + 1\right]^2} \tag{8}$$

where $\tau_n$ is the time that the collaborative edge device is not participating in the collaboration evaluation and $\tau_d$ is decay time interval.

In the CEEC environment, the decay rate of incentive score will be changed due to the change in the completion time of the collaboration trust evaluation task, the basic cooperation reward score, the incentive score consumed by obtaining the computing resources and the cooperation help provided by CEEC, and the duration of alternating between participating in collaboration and negative collaboration.

## 3. Results

This chapter introduces the experimental environment. We verify the effectiveness, accuracy and motivation of the proposed lightweight fuzzy collaborative trust evaluation model (LFCTEM) from different aspects.

### 3.1. Experimental Setup

The operating environment is Windows 10, and the computer uses 2.30 GHz Intel(R) Core(TM) i7-10875H CPU (Intel, Santa Clara, CA, USA) and 16 GB memory. In order to make the experiment closer to the edge computing scenario, this paper uses OMNET++ 5.6.2 to simulate the interaction data of edge devices in a CEEC network and generate experimental data. MATLABR2020b is used to build the fuzzy logic algorithm and test the trust evaluation. In the experiment, we determine the simulation parameters in Table 3 to obtain the optimal model.

**Table 3.** Simulation Parameters.

| Parameters | Description | Value |
|:---:|:---:|:---:|
| $R_b$ | Basic cooperative reward score | 20 |
| $\vartheta_q$ | Baseline quality reward score | 10 |
| $\varphi_1$ | The quality threshold of collaborative trust evaluation task | 0.2 |
| $\varphi_2$ | The quality threshold of collaborative trust evaluation task | 0.5 |
| $\vartheta_e$ | Baseline efficiency reward score | 10 |
| $\tau_{max}$ | The maximum completion time of cooperative trust evaluation task | 5 (min) |
| $\tau_{dmax}$ | The maximum decay time | 48 (h) |
| $\tau_d$ | Decay time interval | 1 (h) |

In the simulation, we simulate 10 randomly distributed edge servers and 500 dynamic heterogeneous edge devices in the CEEC network, with a range of $1000 \times 1000$ m. The trust evaluation range of all edge devices is set from 0 to 1, and a decision threshold is set in this paper. When the trust evaluation value of the edge device is lower than the threshold, the edge device is considered to be in a malicious state. When the trust evaluation value of the edge device is higher than the threshold, the edge device is considered to be in a trusted state.

### 3.2. Experiment and Analysis

3.2.1. Verify the Incentive Negative Decay Mechanism

A collaboration evaluation example is used to observe the acquisition of incentive scores of collaborative edge devices with different positive degrees. In this experiment, we

set the time of a single collaborative trust evaluation task as 4 h. Based on the incentive scores obtained after successful collaboration, the score is 20. Since the collaborative edge device needs to exchange incentive scores for computing resources and collaboration help provided by CEEC, we simply set the consumption as one incentive score per hour and we set the running time of the instance to the maximum decay time $\tau_{dmax}$, 48 h. In order to show the effect of the incentive negative decay mechanism directly and concisely, participating cooperation and negative cooperation are carried out alternately, which shows that the cooperative edge device intermittently participates in the cooperative trust evaluation task. In Figures 7 and 8, the negative collaboration time of the edge device is set as 1 h, 2 h, 3 h, 6 h, 12 h, 16 h and 20 h. The red time period represents when the edge device is participating in the collaboration trust evaluation task, and the blue time period represents when the edge device is in a negative state.

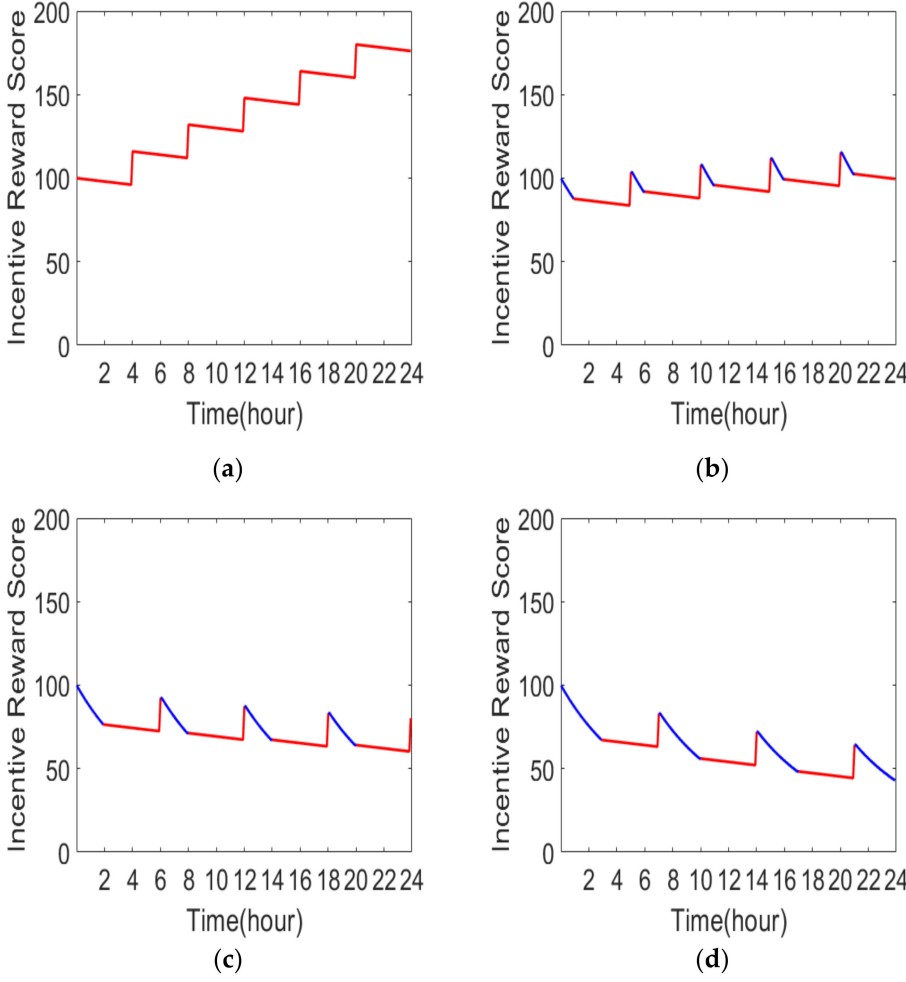

**Figure 7.** Comparison of incentive score with different negative collaboration time(red: positive, blue: negative): (**a**) Edge devices that continuously collaborate actively; (**b**) Edge devices that passively collaborate for 1 h; (**c**) Edge devices that passively collaborate for 2 h; (**d**) Edge devices that passively collaborate for 3 h.

It can be seen from Figure 7a that when the collaborative edge device continuously actively participates in the collaborative trust evaluation task, its incentive score will increase in a stepwise way and a large number of incentive scores can provide the edge device with more computing resources and collaboration help from CEEC. According to Figure 7b,c, when the negative cooperation duration is short (between 0 and 2 h), the edge device can still maintain the reserve of incentive scores by intermittently participating in the cooperative trust evaluation task. According to Figures 7d and 8a, when the negative collaboration

duration gradually increases (between 2 and 6 h), the incentive score obtained by the edge device is no longer enough to compensate for the negative attenuation effect caused by the negative collaboration although the edge device participates in the collaboration trust evaluation task intermittently. At this time, with the continuous decay of the incentive score, the edge device receives the incentive, so it needs to actively participate in the co-operative trust evaluation task to re-enhance the reserve of incentive scores. According to Figures 7d and 8a, when the negative cooperation duration increases to the mid-decay stage (between 6 and 16 h), the incentive score of the edge device will approach 0 with continuous decay. The edge device needs to increase the positivity towards participating in the collaborative trust evaluation task in the mid-decay stage or it will become a negative edge device. It can be seen from Figure 8b–d that when the negative cooperation duration is too long (between 16 and 48 h), the incentive score of the cooperative edge device will decay to 0 within a maximum decay time period (48 h). Furthermore, the edge device will also become a negative edge device and be unable to access the computing resources and collaboration help provided by CEEC.

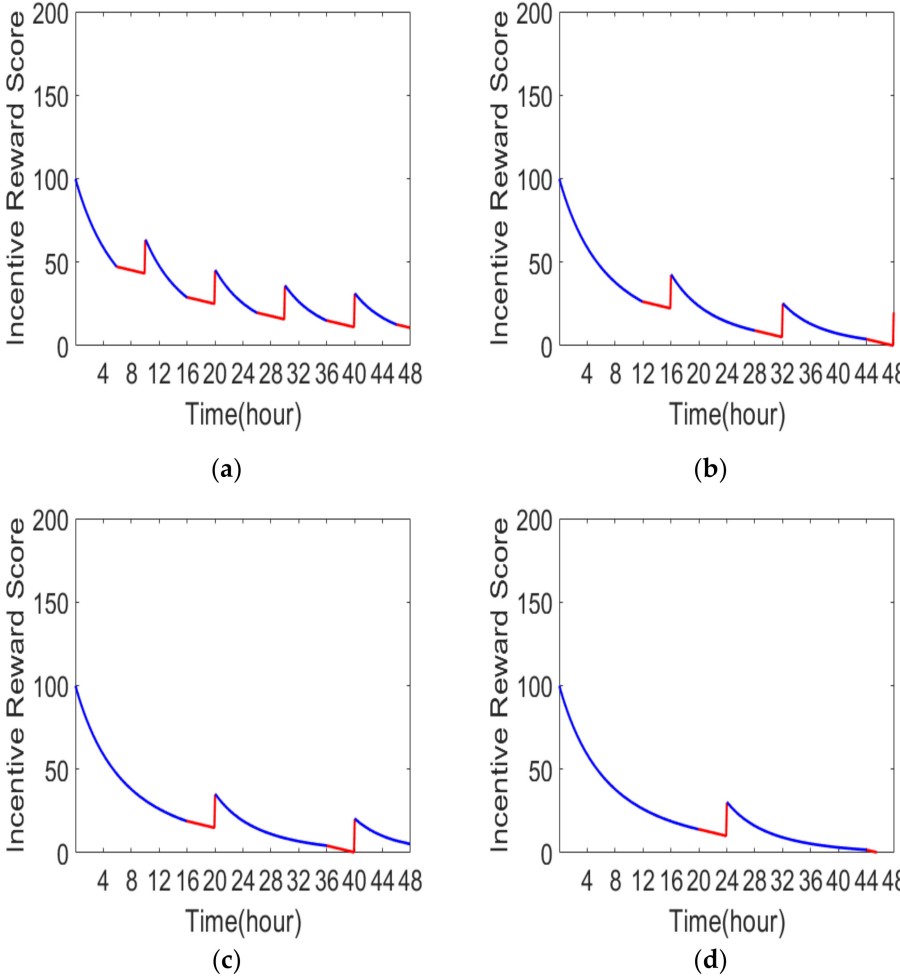

**Figure 8.** Comparison of incentive score with different negative collaboration time(red: positive, blue: negative): (**a**) Edge devices that passively collaborate for 6 h; (**b**) Edge devices that passively collaborate for 12 h; (**c**) Edge devices that passively collaborate for 16 h; (**d**) Edge devices that passively collaborate for 20 h.

### 3.2.2. Verify the Accuracy of LFCTEM by Comparison

Two important indexes for analyzing the accuracy of trust models of edge devices are detection rate and error detection rate of malicious devices. The detection rate is the ratio of the number of detected malicious edge devices to the total number of malicious edge

devices according to the trust evaluation scheme of the model. Error detection rate is the ratio of the number of false detected edge devices according to the trust evaluation scheme of the model (that is, a normal edge device that is mistakenly detected as a malicious edge device, or a malicious edge device that is mistakenly detected as a normal edge device) to the total number of detected edge devices. In this experiment, the detection rate and error detection rate of malicious edge devices in the CEEC network of LFCTEM, DTEM [30] and FITEEV [18] are compared and analyzed at 5%, 10%, 20%, 40% and 50% of malicious edge devices. Among them, DTEM is a trust management system that adds time degradation factor and incentive mechanism. It modifies Bayes' equation by satisfaction function to solve direct trust, and determines the weight of indirect trust value based on improved grey correlation analysis. FITEEV is a reputation data-management system based on fuzzy logic. It introduces packet forwarding rate factor, correctness factor, monitoring factor, speed factor and content change factor to improve the detection rate of malicious devices. Figures 9 and 10 show the detection rate and error detection rate of the three models under different proportions of malicious side devices.

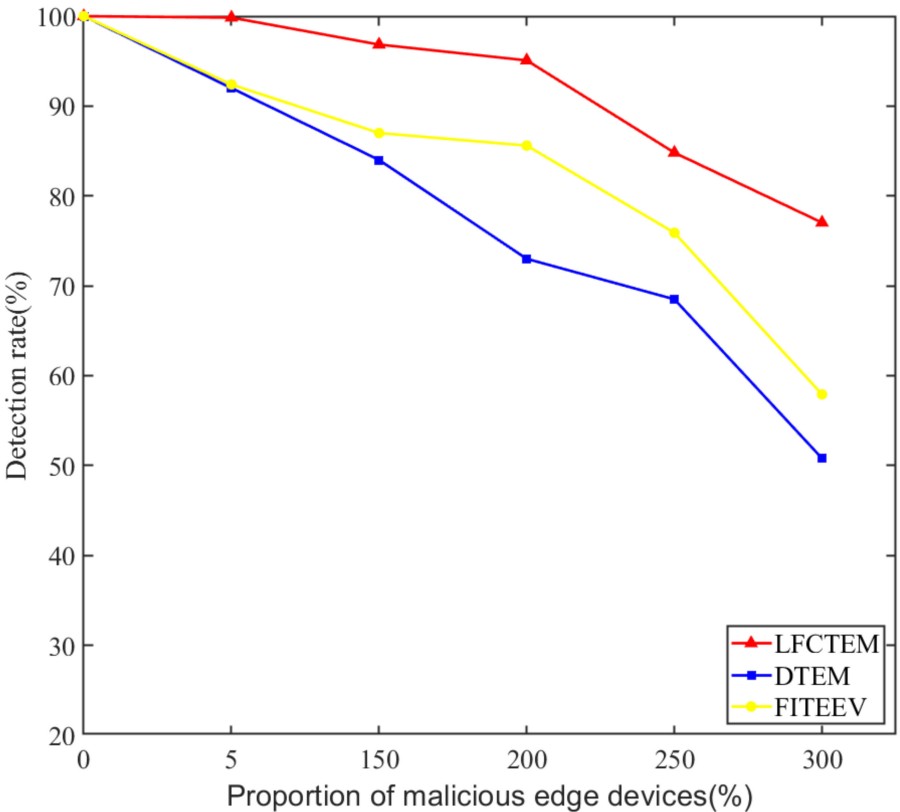

**Figure 9.** Comparison of detection rates of malicious edge devices by LFCTEM, DTEM and FITEEV.

As shown in Figure 9, the detection rate of LFCTEM is significantly higher than that of the other two models, and the detection rates of the first four groups of malicious edge devices with different ratios are all higher than 80%. In the evaluation of trust value, LFCTEM converts the fine-grained trust information of the edge devices group, including success rate of direct interaction, adjacent distance, success rate of public interaction and positivity of cooperation evaluation, into fuzzy trust factors based on the fuzzy logic algorithm so as to further enhance the accuracy of the trust evaluation between the adjacent edges devices, and enhance the malicious edge devices detection rate of the model. As the proportion of malicious edge devices in the CEEC network increases, the detection rates of malicious edge devices of the three algorithms all decrease. However, LFCTEM always keeps a high detection rate of malicious edge devices and the detection rate of malicious

edge devices of LFCTEM increases by 19.11%, so it can be seen that LFCTEM has a better performance than the other two models.

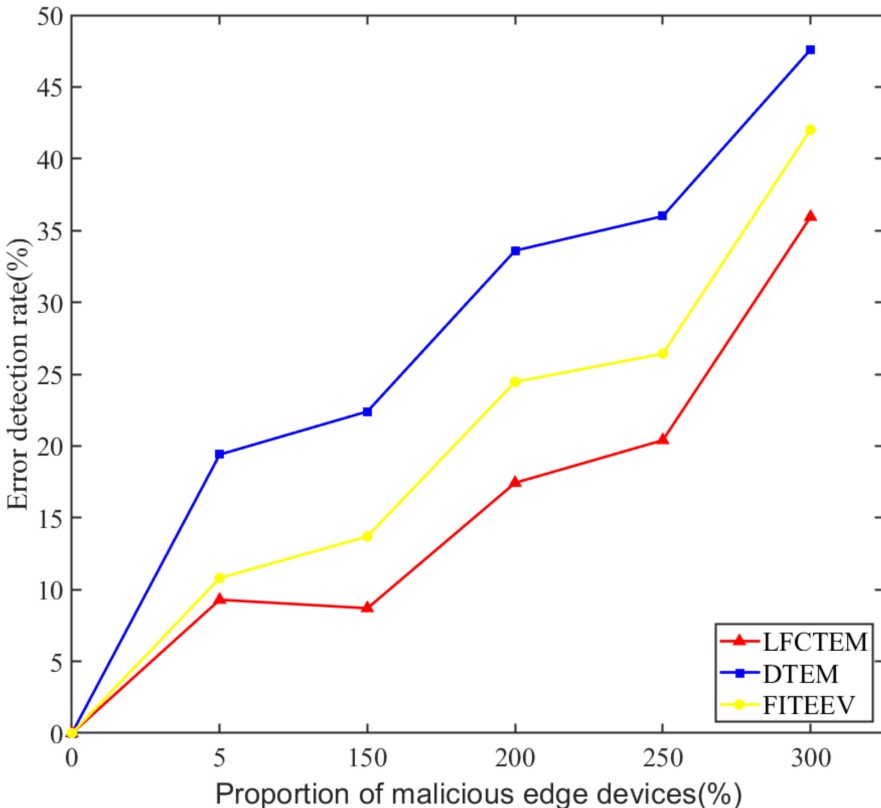

**Figure 10.** Comparison of error detection rate of malicious edge devices of LFCTEM, DTEM and FITEEV.

As shown in Figure 10, the error detection rate increases with the proportion of malicious edge devices. However, the average first four error detection rates of LFCTEM, DTEM and FITEEV models are 13.96%, 27.85% and 18.85%, respectively, and LFCTEM has the lowest false detection rate. Through the analysis, it can be seen that the incentive mechanism is introduced in LFCTEM to reward the accuracy of cooperative trust evaluation of the edge device group using quality reward scores. This is in order to avoid the impact of malicious trust evaluation and false evaluation caused by malicious behavior of edge devices, and, thus, reduce the error detection rate of edge devices. Because FITEEV considers the monitoring factor and the content change factor, it can protect against malicious side devices to a certain extent, and, thus, can guarantee a relatively low error detection rate when the proportion of malicious edge devices is high (40%). However, DTEM does not take corresponding measures, resulting in an error detection rate of 47.60% when the proportion of malicious edge devices reaches 50%, while the error detection rate of malicious edge devices of LFCTEM decreases by 16.20%. As can be seen from the figure, LFCTEM significantly outperforms the other two models.

### 3.2.3. Verify the Anti-Aggression of LFCTEM

We carried out simulation experiments to verify the anti-aggression of LFCTEM. The ON-OFF attack scenario of malicious edge devices is simulated experimentally. The malicious ON-OFF attack is realized by simulating the positive cooperation and negative states of the edge devices alternately. In this attack scenario, a total of 50 rounds of interaction were simulated. Malicious edge devices first cooperate actively in the first 10 rounds of interaction to obtain a high trust value. In rounds 10 to 20 of the interaction, the malicious edge devices launch the ON-OFF attack, changing from the positive cooperative state to the negative state. The attack will damage the CEEC trusted environment. In

rounds 20 to 40 of the interaction, the malicious edge devices continue to pretend to be actively cooperating to cheat trust. In rounds 40 to 50 of the interaction, the malicious edge devices again switch to the negative state for the ON-OFF attack. Figure 11 shows the change of the trust value of the edge devices of LFCTEM in the ON-OFF attack scenario.

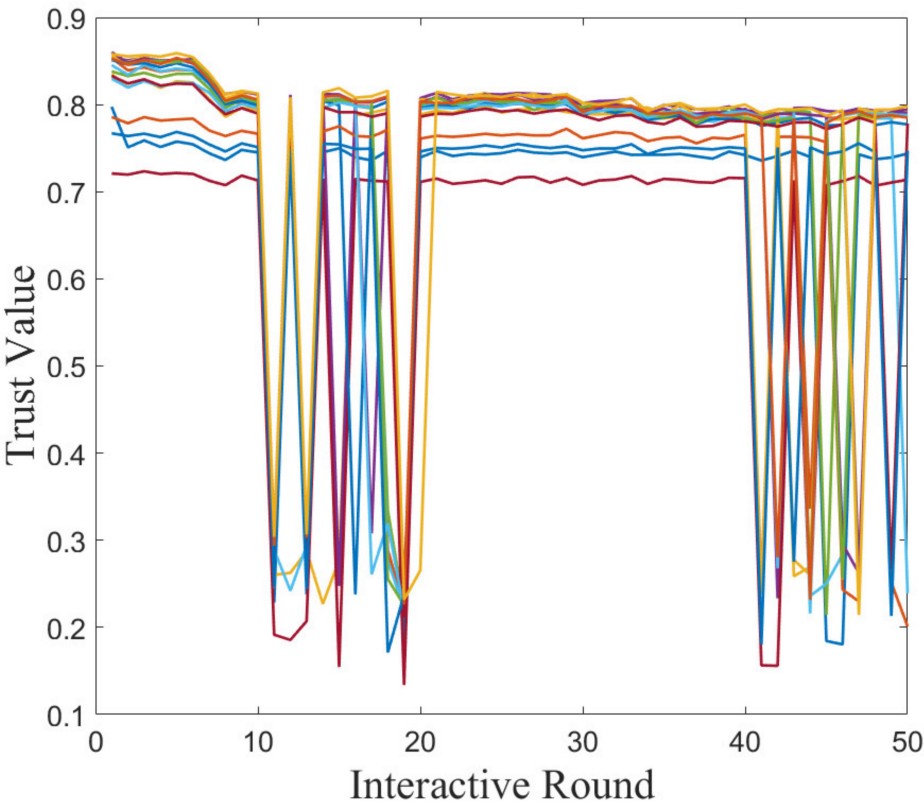

**Figure 11.** Trust value of LFCTEM under ON-OFF attack scenario.

As can be seen from Figure 11, in the initial interaction rounds (the first 10 rounds), the malicious edge devices disguise as a positive cooperation state, so their trust value is in the trusted state. In the interaction rounds from 10 to 20, the malicious edge devices start to launch ON-OFF attacks and switch to the negative state. LFCTEM can quickly and accurately reflect the abnormalities of edge devices through interactive data and the trust values of negative edge devices decrease quickly. Moreover, due to the negative behavior of the edge devices, their incentive scores are greatly attenuated, which leads to the decrease in the CMF factor. The decrease in the CMF factor requires the positive cooperation of the edge devices for a period of time to gradually earn the incentive reward scores. Therefore, the recovery of trust values of malicious edge devices requires better cooperative behavior and more time. This means that the proposed LFCTEM can quickly and accurately identify malicious behaviors and effectively resist ON-OFF attack.

## 4. Conclusions

This paper designs a lightweight fuzzy collaborative trust evaluation model (LFCTEM) for edge devices, aiming at the dynamic and heterogeneous edge computing environment and efficient collaborative scheduling of computing power resources in the CEEC network. In this paper, the success rate of direct interaction, the adjacent distance, the success rate of public interaction and the cooperative incentive score of the group of edge devices in the CEEC network are converted by the membership function by fuzzing the trust factor. We calculate the trust value of the edge devices so as to improve the detection rate of malicious edge devices. Finally, in order to effectively solve the selfish behavior of edge devices, this paper designs a cooperative incentive score mechanism, which improves the positivity of collaborative trust evaluation of edge devices in the CEEC network by constructing

basic incentive scores, quality incentive scores and efficiency incentive scores. We also design the incentive negative decay mechanism to ensure long-term incentives to mitigate the intermittent selfish behavior of the edge devices. Finally, simulation experiments are carried out on the theoretical basis and the feasibility and stimulation of the proposed model in the CEEC network are verified from many aspects. Compared with DTEM and FITEEV, our model enhances the detection rate of malicious edge devices by 19.11%, which improves the reliability of the CEEC trust environment. Furthermore, our model reduces the error detection rate of edge devices by 16.20%, thus alleviating error reporting of the CEEC trust environment.

As CEEC networks expand unceasingly, the efficient coordination scheduling of the vast amounts of highly dynamic heterogeneous computing power resources is still a big challenge. The next steps for research to focus on are how to make full use of the computing power resources, further reduce the idle time, build the multi-layer suitable for the large-scale deep learning trust evaluation scheduling network. In future work, the multi-dimensional trust attribute will be further considered to study the high dynamic adaptability and the high computational power resource utilization of the model.

**Author Contributions:** Conceptualization, C.Y.; methodology, C.Y.; software, C.Y.; validation, C.Y., G.X. and J.C.; formal analysis, C.Y.; investigation, C.Y.; resources, G.X.; data curation, C.Y.; writing—original draft preparation, C.Y.; writing—review and editing, C.Y.; visualization, C.Y.; supervision, G.X.; project administration, C.Y.; funding acquisition, G.X. All authors have read and agreed to the published version of the manuscript.

**Funding:** This research received no external funding.

**Institutional Review Board Statement:** Not applicable.

**Informed Consent Statement:** Not applicable.

**Data Availability Statement:** Not applicable.

**Conflicts of Interest:** The authors declare no conflict of interest.

## Nomenclature

| Parameter | Description |
| --- | --- |
| DISRF | Direct interaction success rate factor |
| ADF | Adjacent distance factor |
| PISRF | Public interaction success rate factor |
| CMF | Cooperative incentive mechanism factor |
| $\varrho_{ij}$ | Success rate of interaction between an edge device and its neighboring edge devices |
| $\{Device_n\}$ | Neighboring edge devices of the edge device |
| $R_r$ | Number of successful interactions |
| $R_e$ | Number of failed interactions |
| $d_{ij}$ | Interaction distance between the host and guest of the edge device group |
| $d_i^x, d_i^y$ | Location of the host edge device |
| $d_j^x, d_j^y$ | Location of the guest edge device |
| $\varrho_{mn}$ | Success rate of interaction between the host and guest of the edge device group |
| $\{Device_k\}$ | Public edge devices for the host and guest of the edge device group |
| TEV | Trust evaluation value |
| $T_{\{user\}}$ | Cooperative trust evaluation task |
| $\{user\}$ | Edge device group |
| $\tau_{max}$ | Maximum completion time |
| $R_b$ | Basic cooperative reward score |
| $R_q$ | Quality reward score |
| $R$ | Total incentive score |
| $R_e$ | Efficiency reward score |
| $\varphi_1, \varphi_2$ | Quality threshold |

| $\vartheta_q$ | Baseline quality reward score |
| $\varphi_q$ | Completion quality of collaborative trust evaluation task |
| $\delta_r$ | Number of times that the collaboration evaluation of the edge device group is correct |
| $\delta_e$ | Number of times that the collaboration evaluation of the edge device group is wrong |
| $\vartheta_e$ | Baseline efficiency reward score |
| $\tau_e$ | Actual time for the cooperative edge device to complete the collaboration trust evaluation task |
| $\tau_d$ | Decay time interval |
| $R_d$ | Incentive negative decay rate |
| $\tau_n$ | Time that the collaborative edge device is not participating in the collaboration evaluation |

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
