# Peer review of "A Fuzzy-Based Co-Incentive Trust Evaluation Scheme for Edge Computing in CEEC Environment"

_applsci, doi:10.3390/app122312453_

Round 1

Reviewer 1 Report

Applied Sciences

Reviewer Report

This is an interesting contribution to the existing literature, but the paper suffers from several shortcomings listed below:

1.     The gap in the previous literature in the field which justifies this study is required to be better explained. The knowledge gap needs to be rewritten and clearly expressed.

2.     Add a separate section to talk about “Cloud-Edge-End 12 Collaboration(CEEC) networks” and the associated trust and security issues.

3.     More explanation should be included in the “Materials and Methods” section in terms of the steps that were followed to conduct the study.

4.     It is recommended to cite more up-to-date literature on the area, especially those published in MDPI journals, for example:

·       Zhang, J.; Lu, C.; Cheng, G.; Guo, T.; Kang, J.; Zhang, X.; Yuan, X.; Yan, X. A Blockchain-Based Trusted Edge Platform in Edge Computing Environment. Sensors 2021, 21, 2126. https://doi.org/10.3390/s21062126

·       Hossain, M. D., Sultana, T., Nguyen, V., Rahman, W. u., Nguyen, T. D. T., Huynh, L. N. T., & Huh, E.-N. (2020). Fuzzy Based Collaborative Task Offloading Scheme in the Densely Deployed Small-Cell Networks with Multi-Access Edge Computing. 10(9), 3115.  https://doi.org/10.3390/app10093115

5.     It is recommended to add the below two sections:

·       Limitations and future research directions

·       Conclusion

6.     The conclusion needs to highlight the significance of the study

7.     There is a need for proofreading and professional editing.

Reviewer 2 Report

In this paper, authors design a lightweight fuzzy collaborative trust evaluation model (LFCTEM) for edge devices, and calculate the trust values of edge devices by fuzzifying trust factors.
Few of the comments are:

It would be better to compare the key achievements of the proposed work with the relevant reported work in the literature in the form of comparison table to better highlight the significance of the proposed work. 

Few of the figures in the paper are blurred. The quality can be improved.  

Reviewer 3 Report

After closely reviewing the entire article, I noticed various flaws. Before publishing, authors should think about the following points:

1-The introduction section has to be condensed.

2-The introduction section also lacks sufficient citations. The authors are suggested to use these five sources and cite them when discussing topics that go beyond the scope of this paper.

https://link.springer.com/article/10.1007/s10586-022-03738-5

https://www.mdpi.com/2076-3417/12/17/8906

https://www.mdpi.com/2079-9292/11/17/2741

https://www.mdpi.com/2079-9292/11/13/1932

https://link.springer.com/article/10.1007/s10586-022-03776-z

3- No section should be left blank. For example, the authors suddenly switched from section 2 to 2. 1. Fill up section 2 with appropriate sentences.

4-If formulas are borrowed from other works, they must be cited.

5- I was curious as to why the authors deleted the conclusion part.

6-The implications of the work beyond the scope must be stated in the conclusion section.

Reviewer 4 Report

Excellent Article.

Include the main intension of the formula presented.

Provide a high-resolution picture.

Round 2

Reviewer 1 Report

Your paper is an interesting contribution to the existing literature in the field 

Reviewer 2 Report

The revision seems good.

Reviewer 3 Report

It can be accepted now.